# Influence of Material Parameters on the Contact Pressure Characteristics of a Multi-Disc Clutch

**DOI:** 10.3390/ma14216391

**Published:** 2021-10-25

**Authors:** Yujian Liu, Man Chen, Liang Yu, Liang Wang, Yuqing Feng

**Affiliations:** 1School of Mechanical Engineering, Beijing Institute of Technology, Beijing 100081, China; liuyj@bit.edu.cn (Y.L.); wangliang@bit.edu.cn (L.W.); fengyq@bit.edu.cn (Y.F.); 2The Key Laboratory of Modem Measurement and Control Technology in Ministry of Education, Beijing 100192, China

**Keywords:** multi-disc clutch, material parameters, contact pressure, temperature field

## Abstract

As an essential part of the transmission, the life of the clutch directly affects the stability of the transmission. In this paper, a finite element model and a thermodynamic numerical model of a multi-disc clutch are established to investigate the influence of material parameters on the contact pressure distribution. The pressure distribution index (PDI) is firstly proposed to evaluate the pressure difference among friction pairs. Moreover, the correctness of the numerical model is verified by the clutch static pressure experiment. The results show that increasing the elastic modulus and Poisson’s ratio of the backplate can effectively improve the uniformity of the contact pressure. However, the variations in material parameters of other clutch components can not easily smooth the pressure difference. Therefore, optimized material parameters for the clutch are proposed, where the maximum pressure and temperature differences are reduced by about 27.2% and 10.3%, respectively.

## 1. Introduction

The wet multi-disc clutch, determining the reliability and safety of the transmission, has always been pivotal in vehicle transmissions [1,2]. To extend transmission life, considerable efforts have been devoted to the failures of wet clutches, such as wear, buckling, cracks, etc. [3]. It should be noted that the friction material plays a substantial role in wear performance. Fei et al. [4] found that the friction components with high porosity could decrease the surface temperature, thus leading to the enhancement of wear resistance. Since Fe could enhance the strength and hardness of material, the wear rate decreased before increasing as the Fe content increased [5]. Yu et al. [6] investigated the wear mechanisms of copper-based and paper-based friction materials, indicating that the wear depth increased dramatically with the ambient temperature increase. Moreover, Zhao et al. [7] verified the dramatic influence of temperature on wear characteristics via pin-on-disc tests. Li et al. [8] and Zhao et al. [9] studied the thermal buckling phenomenon of clutches via sliding experiments and finite element analysis. They suggested that buckling occurred when temperature distribution was non-uniform in the radial direction. Yang et al. [10], Wu et al. [11] and Li et al. [12] investigated the clutch crack phenomenon, suggesting that cracks usually appeared near hotspots. Therefore, clutch stability is crucially affected by the temperature of friction components, and excessive temperature is a major reason for clutch failures.

Many numerical simulations and experimental demonstrations have been conducted in order to improve the clutch temperature distribution. Li et al. [13,14] established clutch heat transfer models, and revealed the influences of the density and specific heat capacity of the carbon fiber on the clutch temperature field. Reduced carbon fiber content led to an increase in porosity, which could discharge lubrication oil effectively, resulting in decreased surface temperature [4]. Yu et al. [15] investigated the temperature of a Cu-based clutch by proposing a thermodynamic model. Wang et al. [16] found that the friction material with the granulated carbon black had lower temperature compared to high-strength graphite. Marklund et al. [17] revealed that friction material permeability could affect clutch temperature distribution. Additionally, many factors can affect the material permeability, such as composition, temperature and operating conditions [18]. Ma et al. [19] proposed that the heat capacity of the metal matrix could also determine its temperature. The wear mechanism of Cu-based brake pads was investigated by Xiao et al. [20]. They found that some debris could be trapped between friction surfaces, then leading to a further increase in temperature. Therefore, almost every aspect of the friction materials has a significant influence on clutch temperature distribution.

In addition, the working conditions are also important in terms of temperature. Zhang et al. [21] found that increasing the working pressure, initial angular velocity and permeability of the friction discs could lead to a significant temperature increase. Mahmud et al. [22] and Kong et al. [23] investigated the clutch temperature fields with consideration for the grooves of friction discs. Zhao et al. [24] simulated clutch non-uniform contact, indicating that the surface temperature gradually increased as the contact ratio decreased. Many works illustrated that the temperature field was also directly determined by the friction materials. 

However, these studies mainly focused on the failures and temperature transfer mechanism with a simplified single friction-pair model. The actual clutch structural characteristics significantly affect the contact pressure distribution difference, thus leading to uneven temperature distributions on friction components [25]. As shown in Figure 1, during the clutch engagement process, the piston together with friction components moves axially by hydraulic pressure, while the circlip in the groove of the cylinder liner restricts their axial movement; gaps between friction components are eliminated progressively, and then the sliding friction occurs. Such a circlip restraint form leads to a dramatic concentration of force in the clutch, resulting in the uneven distribution of contact pressure. Due to the limitations of the spatial arrangement of the transmission system, such a structure is widely used on heavy-duty vehicles. It is known that temperature and pressure fields have a positive correlation [26]. The influence of clutch structure on the contact pressure distribution has already been fully investigated [27,28]. However, the influence of material parameters is rarely investigated, and these possibilities remain to be explored. 

In this paper, a finite element model and a thermodynamic numerical model are developed with consideration for the actual structural characteristics of a multi-disc clutch. Static pressure experiments are conducted to verify the actual contact pressure distribution. Moreover, PDI is employed to evaluate the influence of different materials on the radial pressure distribution. Eventually, the optimized material characteristics and the corresponding temperature fields are obtained. This paper presents a wide range of possibilities for further optimizing the clutch pressure distribution characteristics by material parameters.

## 2. Thermodynamic Model

### 2.1. Contact Pressure Model

In order to study the transmission law of concentrated pressure, a semi-infinite solid cylindrical-coordinate system has been established, as shown in Figure 2.

After applying a concentrated load at the coordinate point A(0, 0, *c*), the following equation can be obtained according to the force balance condition.
(1)$$F_{j}=-\int_{0}^{\infty }2\pi r\sigma_{z}dr\;\left(z>c\right)$$
where *F_j_* is the concentrated force, *r* is radial coordinate, and *σ_z_* is pressure in the *z*-axis direction.

The relationship between the pressure distribution at any point B can be obtained from the Galliakin displacement function as [29]
(2)$$\left\{ \begin{array}{l} \sigma_{r}=\frac{\partial }{\partial z}\cdot \left[\nu \cdot \Delta Z-\frac{\partial^{2}Z}{\partial r^{2}}\right]\\ \sigma_{\theta }=\frac{\partial }{\partial z}\cdot \left[\nu \cdot \Delta Z-\left(1-r\right)\frac{\partial Z}{\partial r}\right]\\ \sigma_{z}=\frac{\partial }{\partial z}\cdot \left[(2-\nu )\Delta Z-\frac{\partial^{2}Z}{\partial z^{2}}\right] \end{array}\right.$$
where Z is the Galliakin function and Δ is the Laplace operator.

When the concentrated pressure is applied at the point *O* (c = 0), the pressure distribution in each direction can be deduced as follows:
(3)$$\begin{array}{l} \sigma_{r}=\frac{F_{j}\left(1-v\right)}{2\pi \left(1-v\right)}\left[\frac{1-2v}{\xi +z\sqrt{\xi }}-\frac{3r^{2}z}{{\left(\sqrt{\xi }\right)}^{5}}\right]\\ \sigma_{\theta }=\frac{F_{j}\left(1-2v\right)}{8\pi (1-v)}\left[\frac{z(2v-1)}{{\left(\sqrt{\xi }\right)}^{3}}+\frac{4(1-v)}{\xi +z\sqrt{\xi }}\right]\\ \sigma_{z}=-\frac{3F_{j}z^{3}}{2\pi {\left(\sqrt{\xi }\right)}^{5}} \end{array}$$
where *v* is Poisson’s ratio, *ξ* = *r*^2^ + *z*^2^.

### 2.2. Thermodynamic Numerical Model

As can be seen from Equation (3), the contact pressure remains the same in the circumferential direction. Thus, the heat transfer equation can be simplified as two-dimensional as follows:
(4)$$\rho c\frac{\partial \psi }{\partial t}=\lambda \left(\frac{\partial^{2}\psi }{\partial r^{2}}+\frac{1}{r}\frac{\partial \psi }{\partial r}+\frac{\partial^{2}\psi }{\partial z^{2}}\right)$$
where *ψ* is the temperature and λ, c and *ρ* are the thermal conductivity, specific heat capacity and density of the friction material, respectively.

The heat flux equation between the frictional pairs is:
(5)q=μ(σ,n,ψ)⋅σ(r,θ,z)⋅n⋅r
where *n* is the relative speed between the friction pairs.

The friction coefficient is obtained from the pin disc experiment as [8]:
(6)$$\begin{array}{cc} \mu & =\frac{0.01\ln\left(4u+1\right)}{e^{0.005\psi }}-\frac{\ln\left(28.3\sigma \right)}{200}+0.035+23e^{\left(\frac{-2.6u}{\left(\ln\psi -3.2\right)\left({\left(28.3\sigma \right)}^{0.4}-0.87\right)}-5.16\right)}\\ & +0.08\left(e^{-0.005T}-1\right)\left(e^{-0.2u}-1\right) \end{array}$$
where *σ* is the contact pressure and *u* is the friction linear velocity.

The heat partition factor between the frictional pairs is expressed as [30]:
(7)$$\gamma =\frac{\sqrt{\lambda_{s}\rho_{s}c_{s}}}{\sqrt{\lambda_{s}\rho_{s}c_{s}}+\sqrt{\lambda_{f}\rho_{f}c_{f}}}$$
where the subscripts s and f represent steel and friction discs, respectively.
(8)$$q_{s}=\gamma \cdot q;\;q_{f}=\left(1-\gamma \right)\cdot q$$


The thermal boundary conditions for the friction pairs are as follows:
(9)$$\begin{array}{l} {\lambda \frac{\partial \psi \left(r,z,t\right)}{\partial r}|}_{r=r_{in}}=+h_{\mathrm{in}}\left[\psi \left(r,z,t\right)-\psi_{e}\right]\\ {\lambda \frac{\partial \psi \left(r,z,t\right)}{\partial r}|}_{r=r_{out}}=-h_{\mathrm{out}}\left[\psi \left(r,z,t\right)-\psi_{e}\right]\\ {\lambda \frac{\partial \psi \left(r,z,t\right)}{\partial z}|}_{z=0}=q_{a}=\gamma \mu \sigma_{a}(r,\theta ,z)\omega r\\ {\lambda \frac{\partial \psi \left(r,z,t\right)}{\partial z}|}_{z=H}=q_{b}=\gamma \mu \sigma_{b}(r,\theta ,z)\omega r\\ \psi \left(r,z,t\right){|}_{t=0}=\psi_{0} \end{array}$$
where *t* presents the time, *ψ*_0_ is the initial temperature, *ψ*_e_ is the environmental temperature, and *H* is the thickness of the friction component. *r*_in_ and *r*_out_ are the inner and outer diameters, respectively; *h*_in_ and *h*_out_ are the convective heat transfer coefficients, respectively; *q*_a_ and *q*_b_, and *σ*_a_ and *σ*_b_ are the heat fluxes and contact pressures at the two friction surfaces of the steel disc, respectively.

## 3. Distribution of the Initial Contact Pressure

As shown in Figure 3, a 6-friction-pair clutch finite element model is established to investigate the influence of material parameters on the contact pressure of the friction pairs. The friction disc consists of a friction core and friction linings, and the latter are bonded on both sides of the former. The piston, backplate, steel disc and circlip are usually made of 65Mn steel in heavy-duty vehicles. The actual parameters commonly used for friction components are shown in Table 1. In addition, the contact pair between the backplate and the steel disc is defined as S_C_, and the others are labeled as S_1_, S_2_, …, S_6_. The steel discs are numbered 1, 2, 3, 4 from the piston side to the circlip side. In operation, the circlip is retained and the loading pressure is 100 kPa on the piston. The initial material and structural parameters used in the simulations are presented in Table 1. In order to reveal the influence of material parameters on the contact pressure distribution, the initial EM and PR of steel are set to 160 GPa and 0.29 in the following simulations.

Figure 4 shows the contact pressure distribution of each friction pair. According to the pressure clouds, it is known that the circlip leads to a progressive increase in the contact pressure along the radial direction. The radial pressure distribution is becoming smoother and smoother from S_6_ to S_1_. Thus, except for S_C_, the most significant and smoothest radial contact pressure differences respectively appear in S_6_ and S_1_, where the pressure differences respectively reach 272 kPa and 93 kPa. The maximum pressure is reduced from 361 kPa in S_C_ to 162 kPa in S_1_, and the minimum pressure is increased from 32 kPa in S_C_ to 69 kPa in S_1_.

Figure 5 shows the radial pressure distribution of the clutch friction pairs under initial material conditions. Due to the difference of radial pressure, it is divided into two parts, namely, the pressure smoothing area A and the pressure concentration area B. In area A, the pressure is less than 75 kPa with little fluctuation. In area B, pressure increases rapidly from 75 kPa to 350 kPa. The pressure distribution index (PDI), *k*_1_ and *k*_2_ (kPa/mm) are employed to evaluate the uniformity of the contact pressure in areas A and B, respectively. The PDI is derived from Equation (10)
(10)$$k_{\left(1,2\right)}=\frac{\bar{r\sigma }-\bar{r}\cdot \bar{\sigma }}{\bar{r^{2}}-{\left(\bar{r}\right)}^{2}}$$


## 4. Effect of Material Parameters on the Contact Pressure

In order to evaluate the influence of material parameters on the clutch pressure distribution, the elastic modulus (EM) and Poisson’s ratio (PR) of the friction components are changed to simulate different materials.

### 4.1. Elastic Modulus

The EMs of the steel discs, backplate and circlip are set to three levels: 160 GPa, 210 GPa and 260 GPa, respectively. Similarly, the EM of friction linings is set to 1600 MPa, 2260 MPa and 2700 MPa. As shown in Figure 6, the variations in the EMs of the steel discs, friction linings and circlip have little effect on the radial pressure distribution. However, the variation in the backplate EM can significantly affect the radial pressure distribution.

With the backplate EM increases, the pressure in the radial area between 115 mm and 125 mm is reduced significantly. To be exact, the maximum pressure is reduced from 361 kPa to 300 kPa in S_C_, whereas the contact pressure in S_1_ is only reduced by 4 kPa. Thus, from S_C_ to S_1_, the maximum pressure reduction rate slowly decreases.

Apart from S_C_, variations in material parameters have the most notable effect at S_6_, and the largest pressure difference also appears at S_6_. Figure 7 illustrates that the changes in backplate EM contribute to the slight pressure decrease in area A. As the backplate EM is increasing, the PDI *k*_1_ are 0.46, 0.45 and 0.46, respectively. However, increasing the backplate EM can substantially smooth the pressure distribution in area B, where the *k*_2_ are 14.56, 12.45 and 11.07, respectively.

### 4.2. Poisson’s Ratio

The PRs of the steel discs, backplate and circlip are respectively set to 0.09, 0.19 and 0.29, and the PRs of the friction linings are 0.19, 0.29 and 0.39. Figure 8c,d illustrate that the changes in the PRs of the circlip and the friction linings have a slight influence on the contact pressure variation. As shown in Figure 8a, varying the PR of the steel discs produces only a weak effect on the pressure values at the inner and outer diameter. As shown in Figure 8b, with the drop in the backplate PR, the pressures in the inner and outer diameter increase radically, by 256 kPa and 214 kPa in S_C_, respectively, while from 95 mm to 115 mm in radial position, the pressure becomes lower and lower.

As shown in Figure 9, when the backplate PR is 0.09, the S_6_ pressure decreases from 575 kPa to 8 kPa in area A; however, the pressure increases from 8 to 288 kPa in area B. As the PR increases, the PDI *k*_1_ are respectively −7.03, −3.04 and 0.46, while *k*_2_ are 22.51, 18.59 and 14.56. Therefore, the pressure distribution in S_6_ becomes more uneven with the increase in the backplate PR.

Similarly, when the PR of steel discs changes to 0.09 and 0.19, the *k*_1_ values in S_6_ are respectively 2.04 and 1.61, and the *k*_2_ values are 11.63 and 13.15. It can be seen that the pressure distribution becomes smoother with the decrease in the PR of steel discs. The effect of material parameters of steel discs on the contact pressure is weaker than that of the backplate.

## 5. Optimization of Material Parameters

As the radial pressure difference of steel disc 4 is the greatest among these steel discs, it has the shortest service life [28]. Consequently, S_6_ is selected to evaluate the influence of material parameters on pressure difference, as shown in Table 2 via PDI.

The smaller the *k*_3_ is, the smaller the radial pressure difference is. Increasing EM and PR of the backplate and reducing PR of the steel discs can reduce the radial pressure difference of S_6_. The optimum operating conditions are achieved when the EM and PR of the backplate are 260 GPa and 0.29, and those of the steel discs are 160 GPa and 0.09. As shown in Figure 10, under the optimized working conditions, the radial pressure differences of S_6_ and S_1_ are reduced to 198 kPa and 63 kPa, respectively. The maximum pressure is reduced by 74 kPa, about 27.2%. The *k*_1_ of S_6_ changes from 0.46 to 2.58, and *k*_2_ is reduced from 14.56 to 8.48, contributing to a significant improvement in radial pressure distribution.

## 6. Experimental Verification

### 6.1. Test Rig

The test rig (Nantong YG132-40), as shown in Figure 11, was employed to verify the clutch contact pressure distribution. More precisely, it was connected with the controller via a hydraulic circuit. The clutch pack was placed on the commodity shelf according to the simulation. Fuji pressure test paper was used to measure the contact pressure distribution in the experiment. The paper type was LW, with a pressure range from 2.5 MPa to 10 MPa. In addition, the FPD8010E analysis software was used to obtain the actual pressure values from the pressure test paper. Meanwhile, to highlight the effect of material parameters, 304# steel and aluminum alloys were used in the following tests. The EM and PR of aluminum alloy and 304# steel were 68.9 GPa and 194.02 GPa, and 0.33 and 0.3, respectively, and the piston pressure was set to 6 MPa.

### 6.2. Test Results and Discussion

Numerous repeated experiments were conducted and the corresponding results were highly consistent. Due to the fact that the pressure values remained unchanged in circumferential directions, a 60° sector area was chosen for analysis. Figure 12a shows the pressure of S_C_, S_6_ and S_1_ in the LW paper for the initial material parameters; the values were extracted as shown in Figure 12b. Taking the area where pressure is greater than 2.5 MPa as the concentration area, the area widths of S_6_, S_4_ and S_1_ were 11.5 mm, 14 mm and 18 mm, respectively. Due to the limitations of the test paper range, the maximum values could not be accurately reflected. Nevertheless, the expansion of the concentration area could also prove that the pressure distribution in S_1_ was far smoother than that in S_6_. According to the circlip restriction, the large concentrated force exacerbated the radial pressure difference on all friction surfaces. Moreover, the radial pressure difference was also obvious even at S_1_. Such a problem seriously affects the service life of the clutch.

The material of the backplate was replaced with the 304# steel and aluminum alloy. Since the PRs of 304# steel and aluminum alloy are almost the same, Figure 13a illustrates that the concentration area expands as the EM increases. As shown in Figure 13b, the concentration areas of 304# steel and aluminum alloy were 11.5 mm and 9.5 mm, respectively. As was found, the pressure increased more evenly in 304# steel. Increasing the backplate EM could significantly reduce the difference in pressure distribution. The result was in great agreement with the simulation. Since the friction did not occur on the backplate, many factors, such as wear, did not need to be considered in the selection of the backplate materials; additionally, changes in backplate materials had little influence on the clutch structure. This indicates that materials with high hardness can be used for the backplate, e.g., ceramic materials, composite materials, or new materials in the near future.

Figure 14a shows the pressure distribution of S_6_ on LW test paper with different circlip materials. As the material parameters of the circlip changed, the pressure curve was almost identical and the pressure distribution remained unchanged. Therefore, changes in the circlip material did not affect the clutch pressure distribution, which is consistent with the simulation results.

From the simulations and experiments above, changes in the backplate material have a much greater effect on the clutch pressure difference than any other components’ materials. When loading the piston pressure, the circlip restriction leads to a dramatic concentration of pressure, which causes the deformation of the backplate. The increases in EM and PR can reduce the deformation, thus contributing to a smoother pressure transfer to the friction pairs. Since the backplate directly contacts with the circlip, the inhomogeneity of the concentrated force on the backplate is greater than that on the steel and friction discs. Thus, changing the material parameters of the backplate is the most efficient approach.

## 7. Temperature Field Comparison

To investigate the clutch temperature field under the optimized conditions, the clutch was operated under long-time slipping conditions: piston pressure 0.1 MPa, ambient temperature 40 °C, relative speed 300 r/min and slipping time 5 s. The contact pressure distribution in Figure 5 was used as the initial pressure in the temperature simulation.

As shown in Figure 15, the temperature at the outer diameter was much greater than that at the inner diameter. The temperatures of steel disc 1 and steel disc 4 were crucially lower than any other friction pairs. This was because no friction motion occurs on the backplate side and the piston side. Therefore, steel disc 3 had the largest radial temperature difference of 79.4°C and a maximum temperature of 135.3 °C.

As shown in Figure 16, there was a significant temperature drop in the optimized condition. Similarly, steel disc 3 had the largest radial temperature difference of 64.5 °C and a maximum steel disc temperature of 121.3 °C, roughly a 10.3% reduction. After changing the material parameters of the backplate and steel discs, the clutch pressure was more evenly distributed. Since the heat flow density has a positive relationship with the contact pressure, the temperature distribution was much more uniform with the increasing uniformity of contact pressure.

## 8. Conclusions

Both the finite element model and thermodynamic numerical model of a wet multi-disc clutch were developed to study the influence of material parameters on the contact pressure and temperature distribution. PDI was firstly proposed to evaluate the pressure difference among friction pairs. Additionally, the clutch static pressure experiments were conducted to verify the above numerical models. Finally, the clutch optimum material parameters were put forward. The results are summarized as follows.
Increasing the EM and PR of the backplate and reducing the PR of the steel discs can dramatically reduce the difference in clutch pressure distribution.The material parameters of the friction linings and circlip have a slight influence on the clutch pressure and temperature distribution.Compared with the initial material conditions, the maximum pressure and temperature differences of the optimized material conditions were reduced by 74 kPa and 14.9 °C, about 27.2% and 10.3%, respectively.


## Figures and Tables

**Figure 1 materials-14-06391-f001:**
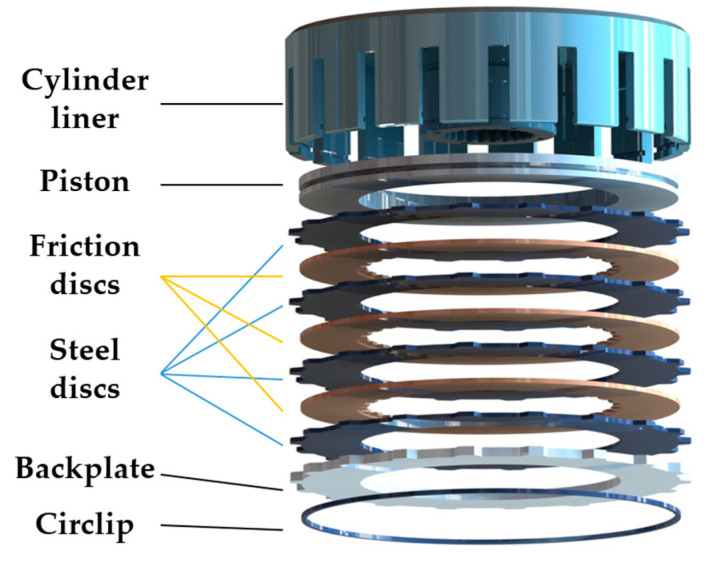
3D diagram of a multi-disc wet clutch.

**Figure 2 materials-14-06391-f002:**
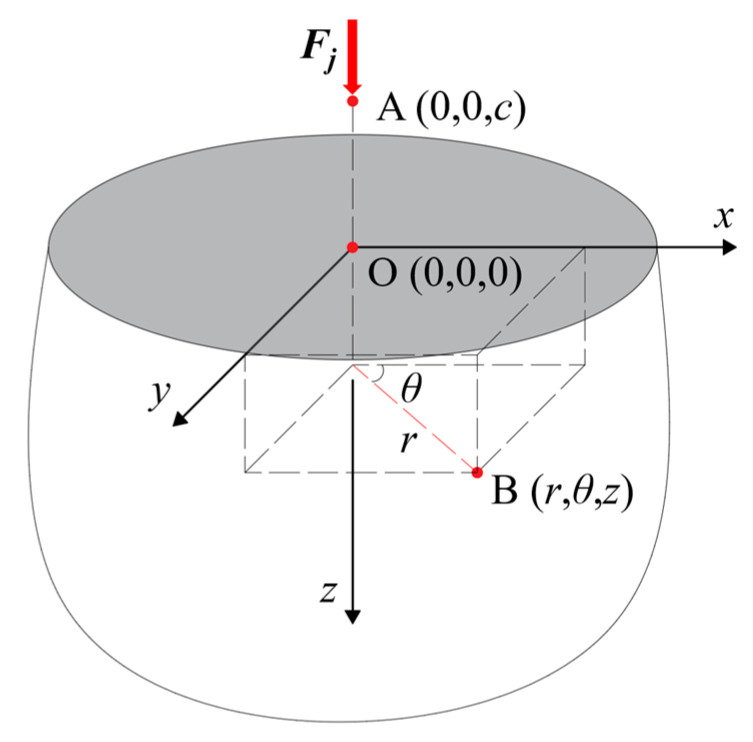
Semi-infinite plate pressure transfer model.

**Figure 3 materials-14-06391-f003:**
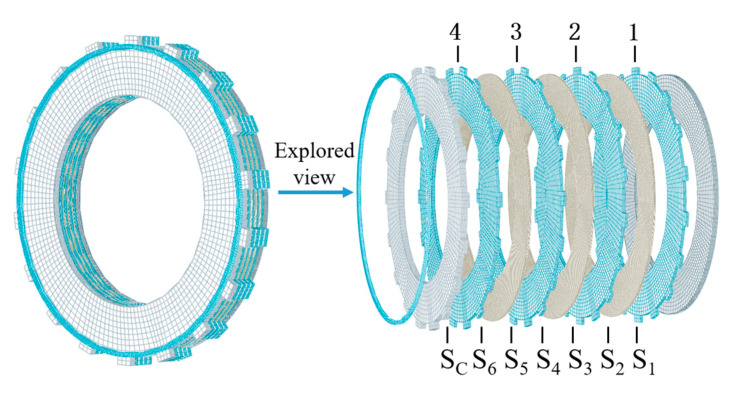
Clutch finite element model.

**Figure 4 materials-14-06391-f004:**
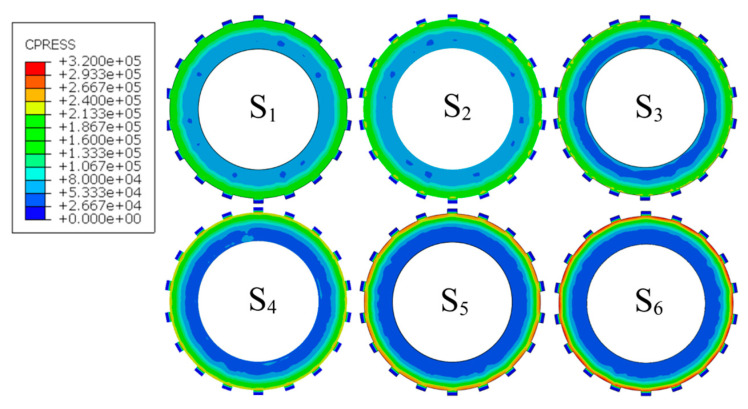
Contact pressure clouds of the clutch pack.

**Figure 5 materials-14-06391-f005:**
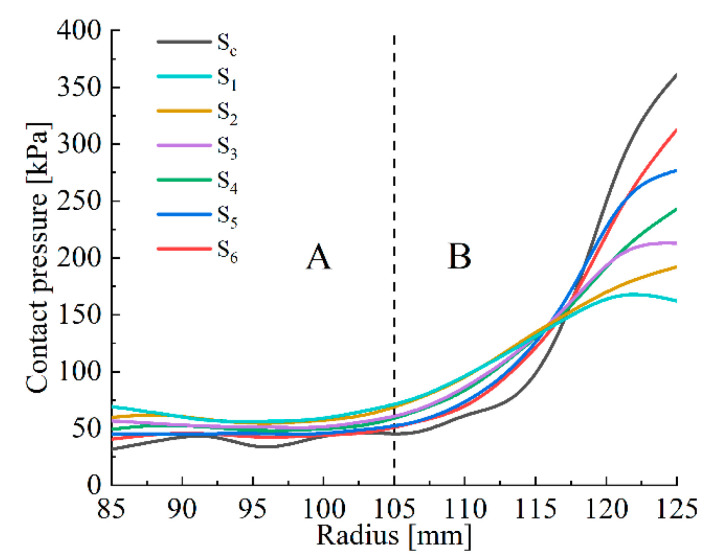
Radial contact pressure distribution of the clutch pack: (A) the pressure smoothing area; (B) the pressure concentration area.

**Figure 6 materials-14-06391-f006:**
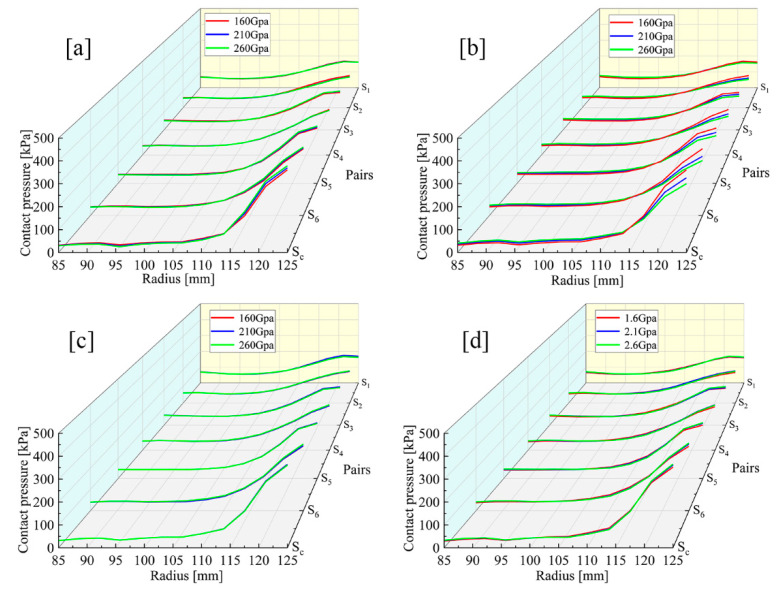
Contact pressure distributions under different EM conditions. (**a**) Steel discs. (**b**) Backplate. (**c**) Circlip. (**d**) Friction linings.

**Figure 7 materials-14-06391-f007:**
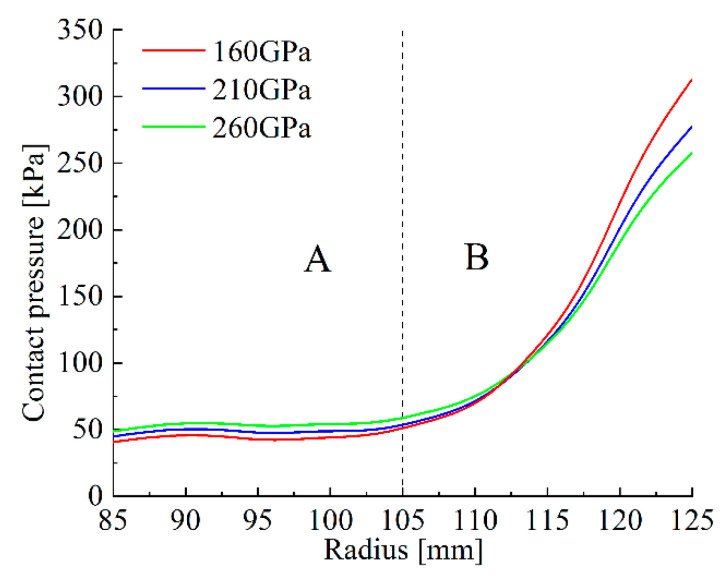
S_6_ pressure distributions under different backplate EM conditions: (A) the pressure smoothing area; (B) the pressure concentration area.

**Figure 8 materials-14-06391-f008:**
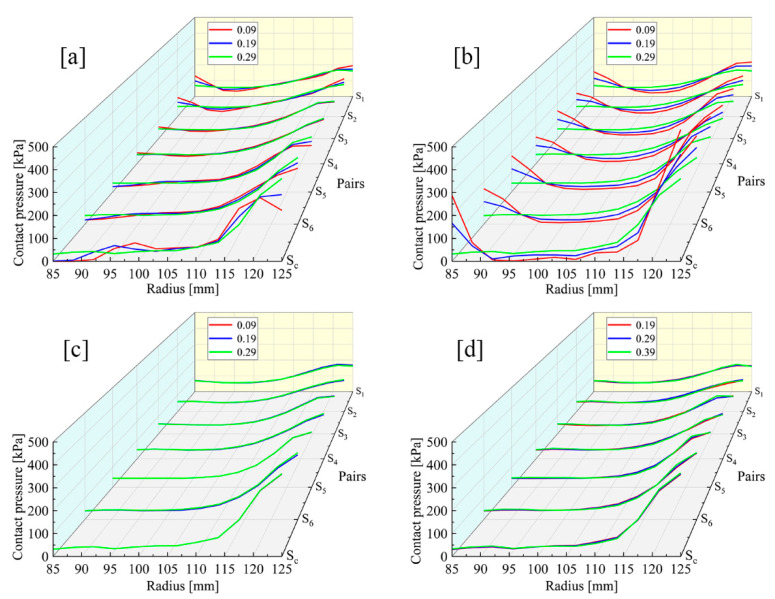
Contact pressure distributions under different PR conditions. (**a**) Steel discs. (**b**) Backplate. (**c**) Circlip. (**d**) Friction linings.

**Figure 9 materials-14-06391-f009:**
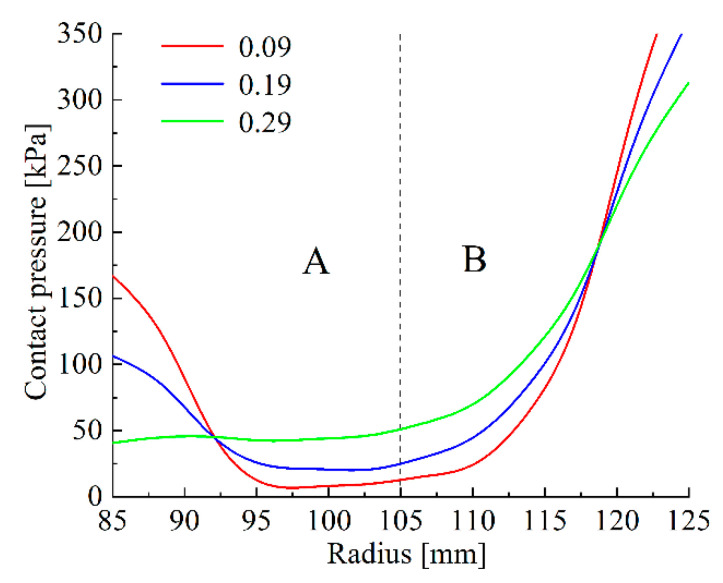
S_6_ pressure distributions of different backplate PRs: (A) the pressure smoothing area; (B) the pressure concentration area.

**Figure 10 materials-14-06391-f010:**
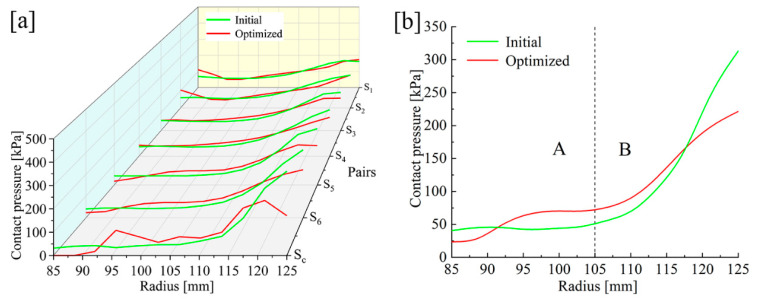
Pressure comparison under initial and optimized conditions. (**a**) Overall comparison. (**b**) S_6_ pressure distribution: (A) the pressure smoothing area; (B) the pressure concentration area.

**Figure 11 materials-14-06391-f011:**
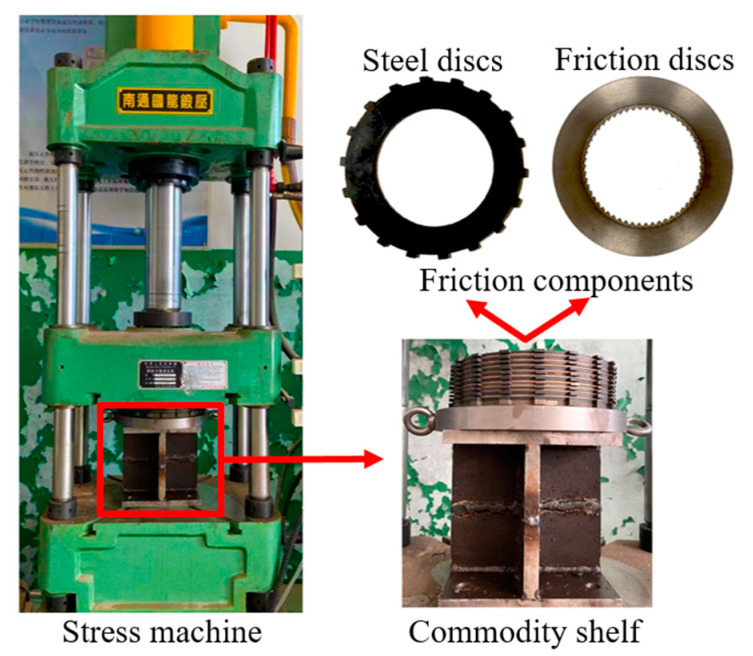
Test rig and samples.

**Figure 12 materials-14-06391-f012:**
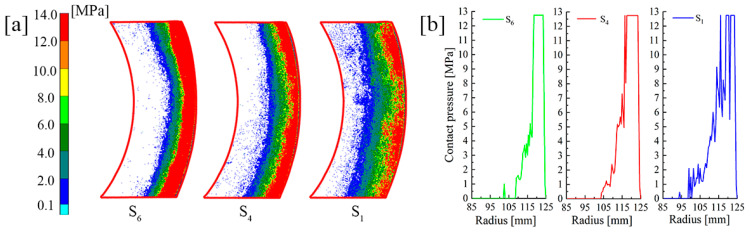
Experimental results under initial working conditions. (**a**) Test paper pressure image. (**b**) Data comparison.

**Figure 13 materials-14-06391-f013:**
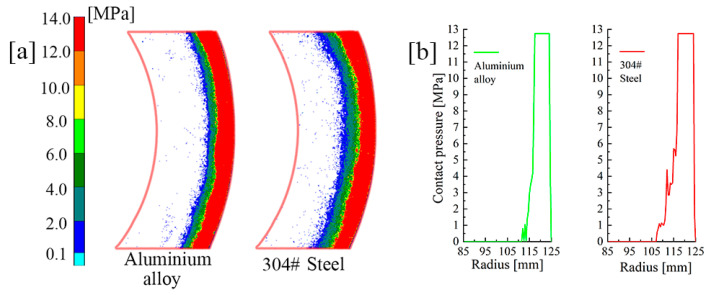
Comparison of tests of different backplate materials. (**a**) Test paper pressure image. (**b**) Data comparison.

**Figure 14 materials-14-06391-f014:**
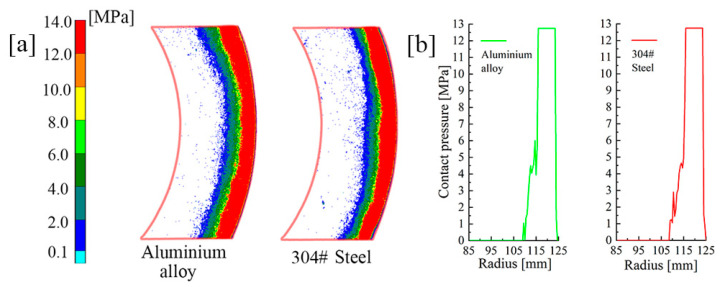
Comparison of tests of different circlip materials. (**a**) Test paper pressure image. (**b**) Data comparison.

**Figure 15 materials-14-06391-f015:**
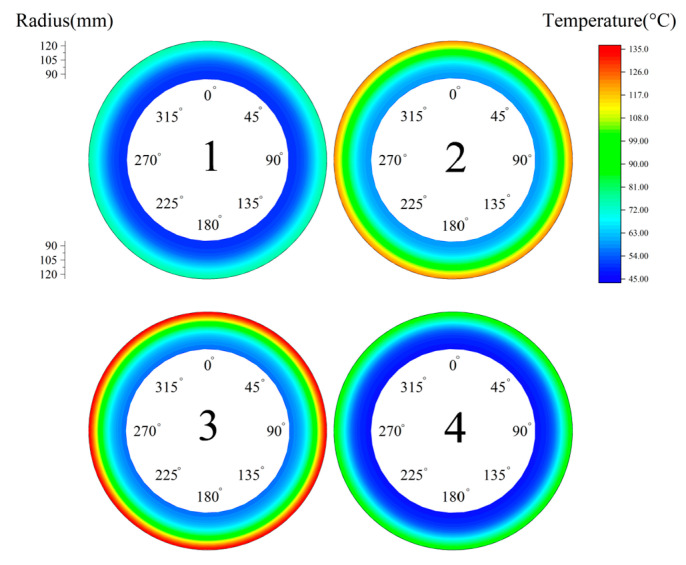
Initial temperature field.

**Figure 16 materials-14-06391-f016:**
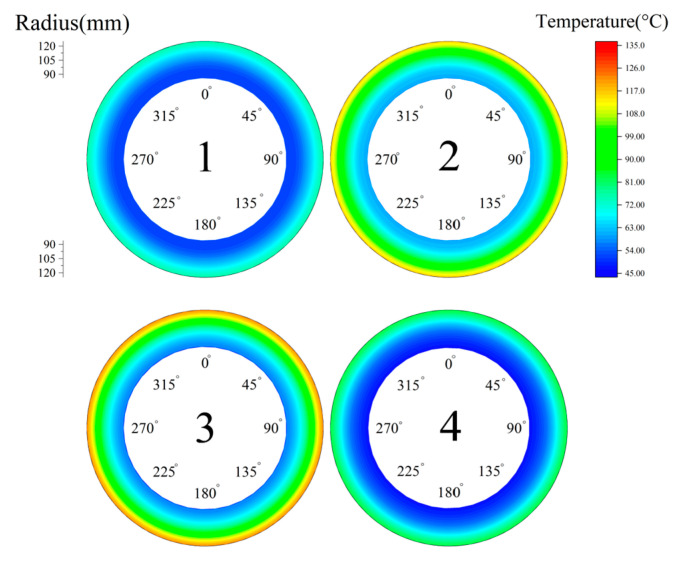
Optimized temperature field.

**Table 1 materials-14-06391-t001:** Initial material and structural parameters of the clutch.

Friction Components	Inner Diameter *r*_in_/mm	Outer Diameter *r*_out_/mm	Thickness *H*/mm	Poisson’s Ratio *v*	Elastic Modulus *E*/GPa
Piston	85	125	6	0.29	210
Steel disc	85	125	3	0.29	210
Friction lining	85	125	0.6	0.27	2.2
Friction core	85	125	2	0.29	210
Backplate	85	125	6	0.29	210
Circlip	122	125	3	0.29	210

**Table 2 materials-14-06391-t002:** Comprehensive evaluation of influencing factors.

PDI				*k* _1_	*k* _2_	*k*_3_ (*k*_1_ + *k*_2_)
Initial group				0.46	14.56	15.02
Material parameters	EM	Backplate	210 GPa	0.45	12.45	12.9
			260 GPa	0.46	11.07	11.53
	PR	Backplate	0.09	−7.03	22.51	29.52
			0.19	−3.04	18.59	21.63
		Steel discs	0.09	2.04	11.63	13.67
			0.19	1.61	13.15	14.76

## Data Availability

The data presented in this study are available on request from the corresponding author.

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
