# Peer review of "Influence of Material Parameters on the Contact Pressure Characteristics of a Multi-Disc Clutch"

_materials, 2021, doi:10.3390/ma14216391_

Round 1

Reviewer 1 Report

The manuscript can be an interesting contribution to the modern industry but In this form, is a more technical report than a scientific paper. There are some issues to improve.
In section 1 Introduction, the authors should dedicate one or two paragraphs to explain the importance of the influence of material parameters on a multi-disc clutch and please review the literature, rather than simply list them. The last paragraph should describe the purpose and method of this paper. The key contribution of this paper should be concentrated and highlighted. I suggest the authors explain better and clearly state the benefits of their research and their result. Clearly describe the novelty of the research and add information about the practical significance of the work. There is no Discussion chapter, why? Chapter numbers are mixed up.
In the present form of the paper, the results are not discussed sufficiently. Please make the findings in conclusion in bullet points to make it easier for readers to catch.

Reviewer 2 Report

In this study, the authors investigated the effects of material properties (elastic modulus and Poisson's ratio) on the contact pressure and temperature distribution of a multi-disc clutch using computational modeling, FEA analysis and subsequent experimental validation. Overall, the method (physical experiment and FEA setup) is clearly illustrated. The results are well presented and the conclusions are supported by the data. 

The quality of this manuscript may be further improved by addressing the comments/concern below:

  1. Missing equation: On page 6, 'The PDI was derived from equation (10).' Where is (10) in the manuscript?
  2. The reviewer understands that the purpose of this manuscript is to investigate the effects of material properties on the pressure distribution and to showcase this, different materials properties will need to be tested and evaluated. However, during this process, the choice of material properties still need justification and the numbers should also be reasonable and/or representatively of the materials properties available for clutch material application.
    1. Specifically, the authors varied the EM to be 160, 210, and 260 GPa, why are these three numbers chosen? Are they representative of common materials used for clutch fabrication? (e.g., 260 GPa seems very high for steel)
    2. In addition, the authors varied the PR to be 0.09, 0.19, 0.29. Again, why are these three levels chosen? Are they representative of common materials used for this application? E.g., a Poisson's ratio of 0.09 for steel disc. A PR of 0.09 for steel is unusually low (Is it even possible to achieve this PR for steel?)
    3. The manuscript concluded with the optimum properties being: EM = 260GPa for backplate and PR = 0.09 for steel discs from the simulation. While this set of optimum value provides some insight for material design, it is also important to provide readers with a realistic context: what is the common values for backplate and steel discs used for wet clutches? Are the EM (260GPa) and PR (0.09) possible for steel?
  3. On page 9, first sentence of section 4, need to include a reference for this claim 'Since the greatest radial pressure difference appears in the steel disc 4, which has the shortest service life of all steel discs'.  In addition, the grammar isn't right for this sentence. 
  4. In the experimental validation part, only one sample was tested for each condition and the results were interpreted from these n=1 data. The author may consider repeating the test, increase the sample size (even just n=3) and run statistical analysis to better support the claim.

Round 2

Reviewer 1 Report

The paper was improved according to my comments.
I find the paper suitable for publication.